# Biomarkers of Liver Injury during Transplantation in an Era of Machine Perfusion

**DOI:** 10.3390/ijms21051578

**Published:** 2020-02-25

**Authors:** Ricky H. Bhogal, Darius F. Mirza, Simon C. Afford, Hynek Mergental

**Affiliations:** 1National Institute for Health Research, Birmingham Biomedical Research Centre, University of Birmingham and University Hospitals Birmingham NHS Foundation Trust, Birmingham B15 2TT, UK; Darius.Mirza@uhb.nhs.uk (D.F.M.); s.c.afford@bham.ac.uk (S.C.A.); 2Centre for Liver and Gastrointestinal Research, Institute of Immunology and Immunotherapy, University of Birmingham, Birmingham B15 2TT, UK; 3The Royal Marsden Hospital NHS Foundation Trust, London SW3 6JJ, UK; 4Liver Unit, Queen Elizabeth Hospital, University Hospitals Birmingham NHS Foundation Trust, Birmingham B15 2TH, UK

**Keywords:** liver transplantation, ischaemia–reperfusion injury, biomarkers, machine perfusion

## Abstract

Liver ischaemia–reperfusion injury (IRI) is an intrinsic part of the transplantation process and damages the parenchymal cells of the liver including hepatocytes, endothelial cells and cholangiocytes. Many biomarkers of IRI have been described over the past two decades that have attempted to quantify the extent of IRI involving different hepatic cellular compartments, with the aim to allow clinicians to predict the suitability of donor livers for transplantation. The advent of machine perfusion has added an additional layer of complexity to this field and has forced researchers to re-evaluate the utility of IRI biomarkers in different machine preservation techniques. In this review, we summarise the current understanding of liver IRI biomarkers and discuss them in the context of machine perfusion.

## 1. Introduction

Liver transplantation (LT) remains the only curative treatment for patients with end-stage liver disease or fulminant hepatic failure. There is a worldwide discrepancy between the number of eligible patients in need of LT and the number of available donor livers. This shortage causes ongoing waiting list mortality [1] and the transplant community has endeavoured to utilise a wider pool of potential donor organs in order to bridge this deficit [2]. Invariably, this has entailed the utilisation of extended criteria grafts, such as steatotic livers and organs donated after circulatory death (DCD). The enthusiasm for using such donors has been fuelled by the increasing experience with machine perfusion (MP) technology [3]. Following the early reports of clinical success with MP for liver transplantation [4], research groups worldwide have used various forms of MP to recondition and optimize different types of donor livers prior to transplantation, in order to increase the donor pool and reduce the waiting list mortality [5,6]. The essential premise of MP is to obviate and mitigate the ischaemia–reperfusion injury (IRI), which is a detrimental part of the transplantation process [7]. Various forms of MP have been developed in an attempt to reduce the IRI magnitude, or to promote donor liver resilience to the detrimental effects of IRI. In this review, we summarise the current understanding of the IRI mechanism and discuss its biomarkers in the context of machine perfusion.

## 2. The Mechanism of the Liver Ischaemia–Reperfusion Injury

The liver IRI was first described by Toledo-Pererya in experiments performed in canine models [8]. This is now understood to be a tissue injury that occurs when the hepatic blood supply is temporarily interrupted and subsequently restored after a variable length of time [9]. The IRI is an intrinsic part of the transplantation process and has detrimental effects on liver functions. In the immediate post-operative phase, it presents as an early allograft dysfunction (EAD), but it also contributes to long-term outcomes when it plays a role in the development of complications such as ischaemic cholangiopathy.

The traditionally accepted model of liver IRI involves a period of cold IRI whilst the donor liver is being procured, followed by a period of warm IRI during the implantation phase of transplantation [10]. It is important to note that much of the basic and translational research performed in this area has involved models of warm IRI, predominantly in small animal experiments [11]. This difference has important implications for biomarker discovery, validation and utility, which is further compounded by the fact that cold and warm IRI most likely have variable effects in different types of donor livers [10]. The IRI encompasses a very complex and broad-based injury that still remains to be completely understood, and it should be considered and assessed within the context of the triggering insult and the type of graft exposed to the injury.

The traditionally accepted model of IRI is undergoing a further paradigm shift with the advent of MP of donor livers and it is likely that in the coming years, the concept of IRI will continue to evolve [5]. We expect that ongoing research will improve the understanding of specific IRI sub-types. In light of this, we discuss here the newly emerging ideas and currently published studies in this field. A simplified overview of the liver IRI is demonstrated in Figure 1.

It is well known that IRI targets and damages all parenchymal cells of the liver, namely hepatocytes, liver sinusoidal endothelial cells (LSEC) and cholangiocytes [12]. During the ischaemic phase of the injury, the lack of substrates and oxygen results in the production of reactive oxygen species (ROS), primarily by the mitochondria, which then induce oxidative stress and perpetuate liver parenchymal injury [13]. Steatotic liver organs produce higher levels of ROS and are, thus, more susceptible to oxidative stress [14]. Paradoxically when blood flow is restored to the liver (i.e., warm IRI) the availability of oxygen further accentuates oxidative stress, increasing the damage to the donor liver [12]. In the reperfusion phase there is a concomitant release of inflammatory cytokines and an influx of an inflammatory cell infiltrate comprising neutrophils and macrophages that amplify tissue injury [15]. The mechanisms are, however, more complex and involve other processes such as autophagy [16], toll-like receptors [17], hypoxia-inducible factor 1 signalling pathways [18] and other cell types, such as hepatic stellate cells and natural killer cells. The liver IRI have been recently reviewed by Dar et al. [10].

The complex nature of the liver IRI can make the evaluation of potential biomarkers difficult, which in turn impacts upon their utility. The ideal biomarker in liver transplantation would predict the extent of graft injury pre-transplant (parenchymal and biliary), and stratify the risk of EAD, primary non-function (PNF) or biliary complications. A single biomarker is unlikely to be able to fulfil these ideal requirements and it is likely that a combination of biomarkers will be required to assess different aspects of donor organ quality. Furthermore, the different types of MP and their individual impacts upon the global IRI might require specific biomarkers for different types of MPs, in the context of LT.

## 3. Overview of Different Techniques of Liver Machine Perfusion

Traditionally, the donor livers immerged in preservation fluid are placed on ice for static cold storage (SCS), after procurement. This organ preservation method remains the standard practice worldwide, which allows the cold IRI to occur. MP attempts to provide vascular flow and oxygen to the donor liver at some point in this process. At present, the liver MP field is broadly divided into two separate areas [5]. The normothermic MP (NMP) involves perfusion of donor livers with blood, or other fluids capable of oxygen carrying capacity, either immediately after procurement or after a period of SCS. NMP maintains an entirely physiological environment and ameliorates the liver ischaemia, with the IRI essentially occurring on the MP device rather than after the liver reperfusion in the recipient [19]. Biomarkers that could predict donor livers at unacceptably high risk of EAD, primary non-function or ischaemic cholangiopathy or other serious post-transplant complications, might, thus, be excluded out of the graft pool, making the transplant procedure safer for the recipients. In the prospective randomised European NMP trial, comparing perfusion with SCS, the perfused arm showed significant improvement in liver utilisation [20]. This was an unexpected finding, as all enrolled livers were from donors that were considered transplantable upon entry into the study, and might have occurred due to the mitigation of IRI that lessened the peri-transplant instability in marginal graft recipients [21].

The hypothermic MP (HMP) involves liver perfusion with cold oxygenated artificial fluids, following a period of SCS, delivered either through the portal vein alone or through the portal vein and hepatic artery combined. The published HMP series results suggest that a relatively short perfusion that last 60–90 min, allows sufficient functional graft recovery prior to implantation, and have demonstrated good outcomes, particularly with respect to biliary complications [22].

Much of the translational and clinical research with NMP and HMP is occurring simultaneously and many groups, such as our own, are working with both techniques. Ultimately, the aim of both perfusion techniques is to prevent or reverse the decline of the donor liver quality caused by SCS and IRI, and to enable a safer utilisation of the extended criteria grafts. Development of the clinical biomarkers of IRI will allow an objective assessment of allograft transplantability and assist in deriving maximal patient benefit.

## 4. Biomarkers of Liver Ischaemia–Reperfusion Injury

The literature contains a large number of markers and molecules which purport to assess the liver IRI [23]. Many have been studied in small animal models involving warm IRI only [24,25,26,27,28] and this might in large part be due to the highly technical nature of small animal models of liver transplantation. Nonetheless these models might provide utility in the use of clinical transplantation, particularly when assessing or predicting the extent of liver damage. Biomarkers might enable the quantification of the extent of parenchymal or biliary injury and assess whether a liver is suitable for transplantation.

The following section discusses potential IRI biomarkers recently detailed within the literature. We caution the reader that many of these biomarkers might be model specific, and have been described in the context of warm ischaemia reperfusion only. A summary of the biomarkers is shown in Table 1.

## 5. MicroRNAs

miRNAs are one class of small noncoding RNAs of 19–24 nucleotides that are important gene regulators at the post-transcriptional level [29]. Many miRNAs seem to be associated with liver IRI, but miRNA-122 (abbreviated to miR-122) shows considerable promise as a biomarker of hepatocyte injury. miR-122 is the most abundant miRNA in the liver, is highly expressed in hepatocytes and accounts for almost 70% of the total hepatic miRNAs [30]. Using a porcine model of ischaemia, Andersen et al. demonstrated that miR-122 is significantly elevated as a result of liver injury [31,32]. These data are further corroborated by liver IRI studies that demonstrate that circulating miR-122 level was significantly increased after 45 min of hepatic ischaemia and 1–24 h of reperfusion [33]. These findings are unsurprising, as IRI targets and damages hepatocytes and are reflective of cellular damage and the pro-inflammatory nature of the injury. Whilst high levels of miR-122 would indicate a significant liver injury, no threshold values are known that suggest irreparable damage. Therefore, this area requires further investigation with subsequent validation in other experimental models and a clinical MP setting.

In a porcine model of DCD liver transplantation coupled with human liver graft biopsies, Li et al. demonstrated that expression of human miR-146b-5p, which is homologous to porcine miR-146b, successfully correlated with the extent of EAD [34]. It is an important distinction as to whether biomarkers can be used to predict both IRI and EAD, which are two different scientific and clinical entities. miR-34 might also serve as a biomarker of liver IRI, but has not been evaluated in the transplant or human setting [35]. Using a variety of acute liver injury models, Schueller et al. demonstrated that miR-223 expression was significantly dysregulated in mice livers after induction of acute liver injury, as well as in liver samples from patients with acute liver failure. In acute and chronic liver injury models, hepatic miR-223 up-regulation was restricted to hepatocytes, and correlated with the degree of liver injury and hepatic cell death. Moreover, elevated miR-223 expression was significantly higher in the serum, following acute liver injury [36].

Whilst miRNA shows promise as biomarkers for hepatocyte injury, and miR-122 in particular has been shown in a variety of acute liver injury models, to date there is insufficient evidence to support the clinical use of miRNA to assess transplantability during MP evaluation.

## 6. Interleukin 17

IL-17 is a pro-inflammatory cytokine, which is secreted by a subset of CD4+ T-cells in response to interleukin 23 and has been linked to the development of autoimmunity. The presence of IL-17 during IRI is likely to be a result of the associated sterile inflammation.

A recent study by Li et al. [37] reported that IL-17 was significantly elevated in mice undergoing warm IRI. The authors also demonstrated a concomitant elevation of the chemokines Eotaxin, Eotaxin-2 and serum tumour necrosis factor receptor 2 during warm IRI. As the liver IRI is a pro-inflammatory process, many of these inflammatory cytokines are involved at different phases of the injury. For instance, one might reasonably expect miRNA-122 to be present early during IRI, as a marker of hepatocyte injury, whereas IL-17 might be detected later in the process when pro-inflammatory cells infiltrate the damaged liver. The dynamic changes make it difficult to predict and interpret how these biomarkers could be used clinically, and the questions can only be answered with further research in relevant experimental or clinical models.

## 7. Interleukin 33 and Cyclin D1

IL-33 is a cytokine that specifically drives the production of Th2-associated cytokines such as IL-4. In the normal liver, LSEC are the primary source of IL-33, but in liver fibrosis IL-33 is found within activated hepatic stellate cells [38]. Transplanting donor livers with severe macrosteatosis is associated with an increased risk of PNF, and in experimental animal models of IRI, hepatic levels of IL-33 decreased while Cyclin D1 levels increased. IL-33 in this context might reflect background liver disease, as opposed to being reflective of liver IRI. Cyclin D1 is a cell cycle protein that is also required for progression through the G1 phase of the cell cycle. Rats with high levels of nuclear Cyclin D1, prior to IRI either did not survive warm IRI or had persistent macrosteatosis, after 7 days on control diet. Cyclin D1 expression also corresponded to delayed graft function [39]. These studies again demonstrate the complexity of biomarker assessment in the context of liver IRI and its relation to the quality of the background liver. In the context of the ongoing shortage of suitable donor livers and fast adoption of MP to the organ preservation and assessment pathways, more research in this area is urgently needed as the number of steatotic donors continue to rise. A nuanced approach to biomarker assessment will allow a more sophisticated approach to donor liver utilisation and recipient matching, which is currently not feasible using the lactate clearance-based transplantability selection criteria [40]. We expect that more insight into the IRI assessment and discovery of surrogate markers for longer-term graft outcomes will transform the liver transplant field over the next decade.

## 8. Fibroblast Growth Factor 21

FGF21 is a hepatokine secreted by the liver that regulates glucose uptake by adipocytes, with its effects being additive to those of insulin. In a recent study involving patients after liver transplantation, serum levels of FGF21 demonstrated that it might reflect the scale of liver IRI as its levels were found to be elevated 20-fold, relative to healthy subjects [41]. Temporal correlation analysis demonstrated a significant association of the peak serum levels of FGF21, at 2 h, with the magnitude of the increase in both serum ALT and AST levels, at 24 h post transplantation [41]. Whilst FGF21 correlates with the levels of liver injury, its ability to predict EAD or later complications is not known. Similarly, there is no matching data regarding measurements of other biomarkers, such as miR-122, in relation to post-transplant outcomes.

## 9. Endothelin-1

ET-1 is a potent vasoconstrictor that is produced by vascular endothelial cells and is one of three isoforms of endothelin. It has long been known that in patients with portal hypertension and cirrhosis, ET-1 is elevated. Whilst the biomarkers discussed above all pertain to primarily hepatocellular injury, there is an urgent need to develop measures to monitor cholangiocyte damage that might be predictive of biliary complications, which remain a significant cause of transplant-related morbidity, particularly in DCD grafts [42]. Long-term, follow-up of recipients suffering from anastomotic biliary strictures showed that on day 7 and at 3 months post-transplant, there were significantly raised serum ET-1 and reduced tumour necrosis factor alpha (TNF-α), and these markers might predict biliary complications [43]. In another study, patients with high serum ET-1 levels within the first post-transplant week were significantly more likely to develop EAD [44]. These preliminary findings, suggesting ET-1 might be a marker of EAD and biliary complications, are yet to be validated in larger transplant series.

## 10. Biomarkers of Ischaemia–Reperfusion Injury during Machine Perfusion

The fluid, circulating through the liver during MP, provides researchers with a unique opportunity to assess potential biomarkers, specific only to the liver, in a real-time setting. Many of the biomarkers detailed above have not yet been assessed and several others have been identified in the machine perfusion setting instead. There is, however, discordance between the markers discovered by basic science and those discovered during MP experiments.

There would be clear clinical benefits in discovering usable biomarkers to assess liver transplantability, but although the experiments are performed on an isolated organ perfusion model, the variability in the quality of the studied livers and the multiple MP types used, adds another layer of complexity. This can be demonstrated by the example of the perfusion fluid, varying from acellular fluids [45] and oxygenated fluids [46], to blood-based perfusates [20].

Whilst the biomarkers discussed above are not exhaustive, it does demonstrate that for a meaningful clinical use it might be necessary to develop panels of biomarkers that are able to quantify hepatocellular injury, EAD, risk of rejection or biliary complications, post-transplantation. Some of the currently utilised biomarkers assessed during MP have been adopted from clinical observations during liver transplantation [40], and the summary of markers tested in the context of liver MP is outlined in Table 1.

In 2011, Guarrera et al. reported that an increase in ICAM-1, IL-8 and TNF-α correlated with IRI in the HMP setting [47]. TNF-α and IL-8 are both proto-typical inflammatory molecules and, thus, their detection following HMP is an intuitive finding, but these would not allow more detailed donor organ stratification. ICAM-1 is an extracellular glycoprotein (also known as CD54) and its shedding into the perfusate would be consistent with cellular and endothelial injury. The same group also reported that increasing levels of perfusate IL-1β and TNF-α were associated with greater IRI [35]. In animal models of HMP, the mRNA expression of NF-κB p65, IL-6 and TNF-α were significantly decreased in the HMP group compared to the SCS samples [48]. A reduction in the pro-inflammatory response is undoubtedly one of the mechanisms involved in the improved clinical outcomes of livers perfused with HMP [22].

Regarding miR-122 as a marker of IRI during MP, correlation was shown with the extent of IRI in preclinical and animal studies [31]. Selten et al. demonstrated that miR-122 levels were associated with liver injury and also correlated with EAD in a porcine NMP model [53]. A recent NMP clinical observation by the Porte group demonstrated the elevated D-dimer levels in livers with poor graft function and high concentrations of D-dimers, early after the start of NMP, thus, can be considered to be a marker of severe IRI and a predictor of poor function [52]. D-Dimers are fibrin degradation products that represent by-products of fibrinolysis, probably in the MP setting, resulting from microcirculatory disturbance in the donor liver, and might reflect the sub-optimal nature of such allografts. Of note, in this study the livers were exposed to longer periods of cold IRI, exacerbating the injury to the liver microcirculation.

Linares-Cervantes et al. described four biomarkers in an NMP setting relating to biliary and hepatocellular viability in an experimental model with different degrees of ischaemia [49]. The authors concluded that a lactate level <2 mmol/L at 4 h of perfusion, increase in urea level ≥0.5-fold, bile/perfusate glucose ratio ≤0.7 and bile/perfusate Na+ ratio ≥1.1 within 4 h of perfusion, strongly correlated with successful liver transplantation after NMP. This study closely replicates our own group’s viability criteria, where lactate clearance is part of a composite scoring system predicting graft utility for transplant [40]. There is an emerging consensus in the NMP field that lactate clearance reflects the suitable global function of a donor liver for clinical transplantation, and might predict early satisfactory function of allograft [54,55,56].

As alluded to above, the development of ischaemic cholangiopathy is a risk in livers from DCD donors, and many teams have investigated biomarkers of biliary function during NMP. In a study by Linares-Cervantes et al., the authors suggested that a bile to perfusate glucose ratio <0.6717 and a bile glucose level <15 mmol/L resulted in 0% incidence of ischaemic cholangiopathy, at long-term follow-up [49]. In addition, this study found 2 new biomarkers (urea and bile Na^+^) to assess hepatocellular and cholangiocyte viability [52]. Furthermore, Lui et al. had previously reported that bicarbonate was detected at higher levels in bile, from functioning grafts during NMP [57]. Op den Dries et al. confirmed that biliary bicarbonate concentration and pH reflect biliary epithelial function being significantly higher (7.63 ± compared with SCS-preserved livers), in combination with lower gamma-glutamyltransferase and lactate dehydrogenase bile concentrations in NMP-preserved livers, compared to SCS [50]. The lower biliary injury in livers that produce bile with higher bicarbonate and pH and lower glucose concentrations, suggest that these parameters might be accurate biomarkers of bile duct integrity during ex situ NMP [56].

More recent data have demonstrated improving sophistication with the development of biomarkers and has built upon the above studies. It is well known that the mitochondria are the essential energy producer for the cell and have a critical role in both the generation of oxidative stress and the recovery from it, following IRI. The precise role of the mitochondria is beyond the remit of this review and was covered in a recent comprehensive review by Hu et al. [58]. The mitochondria is integrally involved in the production of ATP. The assessment of ATP production during MP can be used as a surrogate for graft function, particularly during HMP, as the accumulated succinate is used to re-synthesise ATP as the donor liver recovers from the injury [59]. Our group’s data also demonstrated that cold-to-warm perfusion increased ATP levels within allografts [60]. The latter was determined experimentally by using liver tissue homogenates processed after the cessation of perfusion, and thus, the assessment of ATP was not contemporaneous and at present cannot be used as a biomarker of graft function. The Dutkowski group have demonstrated that real-time analysis of mitochondrial flavin in the perfusate during HMP, correlated with post-transplant allograft dysfunction and early graft loss (AUC 0.79 and 0.93, respectively) and also mirrored changes in lactate clearance and coagulation factors production [51]. Results such as these await clinical validation, but these show it might be possible to assess liver viability in real-time, during hypothermic MP without a dependence on the conventional functional parameters.

## 11. Conclusions

The development of clinically relevant biomarkers of liver IRI, EAD and biliary complications during MP continues to evolve. The perfusion process allows for real-time biomarker assessment that further increases their relevance and translation to clinical use. It is likely that a combination of perfusate and bile biomarkers will allow a superior assessment of global function of donor livers. The challenge for the transplant community is to interpret what the biomarkers mean for particular donor liver utilisation and how it might predict the immediate and long-term post-transplant outcomes of the graft.

## Figures and Tables

**Figure 1 ijms-21-01578-f001:**
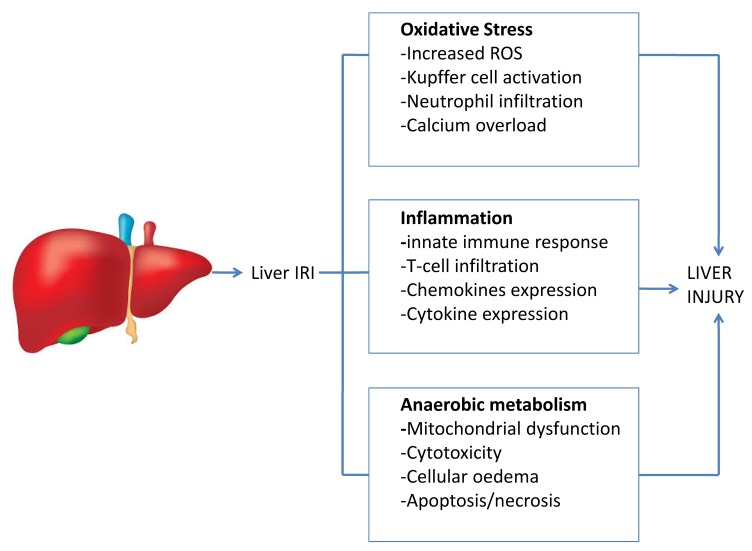
Liver ischaemia reperfusion injury. The scheme demonstrates the multifactorial nature of ischaemia–reperfusion injury (IRI) with multiple processes involved, which ultimately result in liver damage.

**Table 1 ijms-21-01578-t001:** Summary of pre-clinical, transplant-related and machine perfusion-related biomarkers of the liver ischaemia reperfusion injury.

Biomarker	Experimental Model Used	Validated in Transplant	Validated in MP	Injured Assessed		Reference
				IRI/EAD	Biliary	
miRNA						
miRNA-122	Porcine	DCD transplant	No	√	X	[29] *
miRNA46-5p	Human	DCD transplant	No	√	X	[34]
miRNA-34	Rodent	No	No	√	X	[35]
miRNA-223	Murine	No	No	√	X	[36]
Interleukins						
IL-b	Human	Yes	Yes	√	X	[47]
IL-6	Rodent	Yes	No	√	X	[48]
IL-8	Human	Yes	Yes	√	X	[47]
IL-17	Murine	No	No	√	X	[37]
IL-33	Human/Murine	No	No	√	X	[38]
Cytokines and Chemokines						
Eotaxin	Murine	No	No	√	X	[37]
Eotaxin-2	Murine	No	No	√	X	[37]
TNF-α	Human	Yes	Yes	X	√	[43]
Metabolites						
Lactate	Human	Yes	Yes	√	X	[40]
Urea	Human	Yes	Yes	X	√	[49]
Bile/glucose ratio	Human	Yes	Yes	X	√	[49]
Lactate dehydrogenase	Human	Yes	Yes	X	√	[50]
Mitochondrial Flavin	Human	Yes	Yes	√	X	[51]
Miscellaneous						
D-Dimers	Human	Yes	Yes	√	X	[52]
Endothelin-1	Human	Yes	Yes	√	√	[43]
FGF21	Human	Yes	No	√	X	[41]

**Note:** √ designates yes, X designates no, * other references include Lagos-Quintana et al. [30], Andersson et al. [31], Roderburg et al. [32] and Yang et al. [33].

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
