# Peer review of "Biomarkers of Liver Injury during Transplantation in an Era of Machine Perfusion"

_ijms, 2020, doi:10.3390/ijms21051578_

Round 1

Reviewer 1 Report

An excellent review.  The data regarding biomarkers in MP for liver allografts still an emerging area.  The authors have provided an important review to assess the state of the field and correlation of these biomarkers with the mechanisms of liver I/R injury

Author Response

We would like to thank the reviewer for a positive feedback.

Reviewer 2 Report

This review article presented a concise summary of ischemia-reperfusion injury (IRI), biomarkers for IRI and machine perfusion in liver transplantation. Several basic concepts in this field are illustrated briefly including contrast between static cold storage and mechanical perfusion, difference of normothermal and hypothermal mechanical perfusion, mechanisms of ischemia-reperfusion injury and its biomarkers.

Major points:

Number of references are too small. The authors should present evidences for each statement in the description. For example, in line 51, the authors should present past papers reporting ‘the traditionally accepted model of liver IRI’; in line 52, ‘warm IRI’; in line 56-57, ‘the fact that cold and warm IRI are most likely having variable effects in different types of donor livers’; in line 60, ‘a further paradigm’. The authors should carefully revise the manuscript and and cite references for each statement.

Minor points:

In line 64, font of the phrase ‘were found to be’ should be corrected.

Author Response

We would like to thank the reviewer for the constructive and positive feedback. We have followed the recommendation and the revised manuscript version now included 60 references, with most additions in paragraphs highlighted in the reviewer's comment.

The font in the sentence was corrected.

Round 2

Reviewer 2 Report

The authors added substantial number of references to the original manuscript, which help nonexpertized readers with undertstanding present status and future work in this field.